**Data Availability Statement:** All relevant data are within the manuscript and its Supporting Information files.

**Funding:** This work was supported by National Nature Science Foundation of China [grant

# Risk factors for predicting mortality of COVID-19 patients: A systematic review and meta-analysis

**Lan Yang**[⊕][◉], **Jing Jin**[◉], **Wenxin Luo, Yuncui Gan, Bojiang Chen**[‡]*, **Weimin Li**[‡]*

Department of Respiratory and Critical Care Medicine, West China Medical School/West China Hospital, Sichuan University, Chengdu, Sichuan, China

◉ These authors contributed equally to this work.
‡ These authors also contributed equally to this work.
* weimi003@yahoo.com (WL); Cjhcbj@gmail.com (BC)

## Abstract

### Background

Early and accurate prognosis prediction of the patients was urgently warranted due to the widespread popularity of COVID-19. We performed a meta-analysis aimed at comprehensively summarizing the clinical characteristics and laboratory abnormalities correlated with increased risk of mortality in COVID-19 patients.

### Methods

PubMed, Scopus, Web of Science, and Embase were systematically searched for studies considering the relationship between COVID-19 and mortality up to 4 June 2020. Data were extracted including clinical characteristics and laboratory examination.

### Results

Thirty-one studies involving 9407 COVID-19 patients were included. Dyspnea (OR = 4.52, 95%CI [3.15, 6.48], P < 0.001), chest tightness (OR = 2.50, 95%CI [1.78, 3.52], P<0.001), hemoptysis (OR = 2.00, 95%CI [1.02, 3.93], P = 0.045), expectoration (OR = 1.52, 95%CI [1.17, 1.97], P = 0.002) and fatigue (OR = 1.27, 95%CI [1.09, 1.48], P = 0.003) were significantly related to increased risk of mortality in COVID-19 patients. Furthermore, increased pretreatment absolute leukocyte count (OR = 11.11, 95%CI [6.85,18.03], P<0.001) and decreased pretreatment absolute lymphocyte count (OR = 9.83, 95%CI [6.72, 14.38], P<0.001) were also associated with increased mortality of COVID-19. We also compared the mean value of them between survivors and non-survivors, and found that non-survivors showed significantly raise in pretreatment absolute leukocyte count (WMD: 3.27×10⁹/L, 95%CI [2.34, 4.21], P<0.001) and reduction in pretreatment absolute lymphocyte count (WMD = -0.39×10⁹/L, 95%CI [-0.46, -0.33], P<0.001) compared with survivors. The results of pretreatment lactate dehydrogenase (LDH), procalcitonin (PCT), D-Dimer and ferritin showed the similar trend with pretreatment absolute leukocyte count.

numbers 91859203 and 81871890] and Major
Science and Technology Innovation Project of
Chengdu City [grant number 2020-YF08-00080-
GX].

**Competing interests:** The authors have declared
that no competing interests exist.

**Abbreviations:** COVID-19, coronavirus disease
2019; SARS-CoV-2, severe acute respiratory
syndrome coronavirus 2; IQR, interquartile range;
SARS, severe acute respiratory syndrome; CI,
confidence interval; LDH, lactate dehydrogenase;
PCT, procalcitonin.

## Conclusions

Among the common symptoms of COVID-19 infections, fatigue, expectoration, hemoptysis, dyspnea and chest tightness were independent predictors of death. As for laboratory examinations, significantly increased pretreatment absolute leukocytosis count, LDH, PCT, D-Dimer and ferritin, and decreased pretreatment absolute lymphocyte count were found in non-survivors, which also have an unbeneficial impact on mortality among COVID-19 patients. Motoring these indicators during the hospitalization plays a very important role in predicting the prognosis of patients.

## Introduction

Since December 2019, the pandemic of severe acute respiratory syndrome coronavirus 2 (SARS-CoV-2) pneumonia has caused more than 8 million infections, with 440,290 deaths worldwide until June 17th, 2020 [1]. Increasing evidence is investigating the clinical features and laboratory abnormalities in patients with COVID-19 infection. Considering the wide-spread of COVID-19, early and accurate prognosis prediction is urgently warranted. However, the specific symptoms and laboratory biomarkers which may help predict the poor prognosis of COVID-19 patients were unclear. Therefore, we aimed to perform a systematic meta-analysis to summarize the clinical characteristics and laboratory test before treatment among COVID-19 patients and identify the possible risk factors for mortality.

## Materials and methods

### Search strategy

We conducted a systematic search in PubMed, Scopus, Web of Science and Embase to identify studies in patients with COVID-19 infection up to 4 June 2020. The following keywords were used: "2019 novel coronavirus disease", "severe acute respiratory syndrome coronavirus 2", "COVID-19", "2019-nCoV", "SARS-CoV-2" and "clinical", "laboratory", "risk factor", and "mortality", "mortal", "fatality", "fatal", "lethality" or "death". No restrictions on publication status were imposed. Only studies published in English and Chinese were retrieved for this meta-analysis. In addition, reference lists of relevant records were manually screened for further potentially eligible articles.

### Inclusion criteria and exclusion criteria

Two researchers reviewed all articles independently based on titles and abstracts. The inclusion criteria were as follows: 1) all patients were confirmed with COVID-19; 2) studies reported the clinical characteristics, hematological and serological abnormalities both in survivors and non-survivors.

The exclusion criteria were as follows: 1) reviews, letters, case reports, conference abstracts and duplicated publications; 2) insufficient data were provided for extrapolating the mean±SD for hematologic parameters.

### Data extraction and assessment of risk of bias

Studies that met the inclusion and exclusion criteria underwent full-text rescreening. Data extraction was performed by two investigators independently. The following data were

collected: the name of first author, publication year, region of studies, number of the patients with COVID-19, clinical characteristics together with of the laboratory examination in each group. Continuous data were extracted as mean ± standard deviation (SD). While data were expressed as median, range and/or interquartile range (IQR), mean and SD were extrapolated according to Wan et al. [2]. Any disagreements were resolved via discussion and consensus. The risk of bias of each included study was assessed by utilizing the MINORS score [3].

## Statistical analysis

ORs together with the weighted mean difference (WMD) and the 95% confidence interval (CI) were merged and we assessed heterogeneity by using Cochran's Q statistic test and the $I^2$ statistic. When p-values for heterogeneity were no greater than 0.05 or $I^2$ value exceeded 50%, random-model was applied. Otherwise, the fixed-effects model was adopted. We explored the publication bias by the Egger's regression test and the funnel plot. All statistical analyses were conducted by Review Manager (version 5.3), and R (version 3.6.1). Two-tailed P values ≤0.05 were considered statistically significant.

## Results

### Literature search and assessment of risk of bias

A total of 3093 potentially relevant publications were yielded according to our search strategy from PubMed, Scopus, Web of Science and Embase up to 4 June 2020. One additional relevant study was identified from the reference list of included articles. We discarded 1177 articles as duplicates. Two researchers reviewed 1917 articles based on titles and abstracts. After 1839 irrelevant records were excluded, we screened the full text versions of the remaining 78 articles. The following studies were eliminated: reviews, meta-analyses or case reports and studies lacking sufficient data for further analysis. Ultimately, thirty-one qualified articles [4–34] were included in this meta-analysis. The detailed process of the literature search was presented in **Fig 1**. All included studies were non-randomized. The MINORS scores varied between 18 and 21, suggesting a low risk of bias overall (**Table 1** **and S1 Table**).

### Characteristics of included studies

As shown in **Table 1**, the included studies were carried out in China (n = 27), Spain (n = 1), Italy (n = 1), Iran (n = 1) and Poland (n = 1). In total, 9407 confirmed COVID-19 patients were included, of which 7856 were survivors and 1551 were non-survivors. The mean or median age of survivors varied from 40 to 69 years, and that of the non-survivors ranged between 63 to 75.3 years. The proportions of male patients in survivors and non-survivors were 52% and 65%, respectively. For comorbidities, similar to the findings of Ielapi N et al. [35], a history of hypertension was more common among non-survivors (52%) than among survivors (29%). Similar to hypertension, non-survivors were more likely to report having diabetes, malignancy, chronic obstructive pulmonary disease, chronic cardiac disease, cerebrovascular disease and chronic renal disease (**Table 1**). Clinical characteristics included fever, cough, dyspnea, fatigue, diarrhea, myalgia, expectoration, headache, emesis, pharyngalgia, anorexia, abdominal pain, dizziness, hemoptysis, nausea, chest pain, chest tightness and shiver. As for laboratory test, we focused on leukocytes, lymphocytes, procalcitonin (PCT), D-Dimer, lactate dehydrogenase (LDH) and ferritin. Of these studies, nineteen studies provided the clinical characteristics and the laboratory findings of COVID-19 patients [6–9, 11–13, 15, 17, 20, 22, 24, 26–28, 31–34], seven studies only targeted the clinical characteristics [4, 5, 14, 16, 18, 23, 29], and another five studies only focused on the laboratory findings [10, 19, 21, 25, 30].

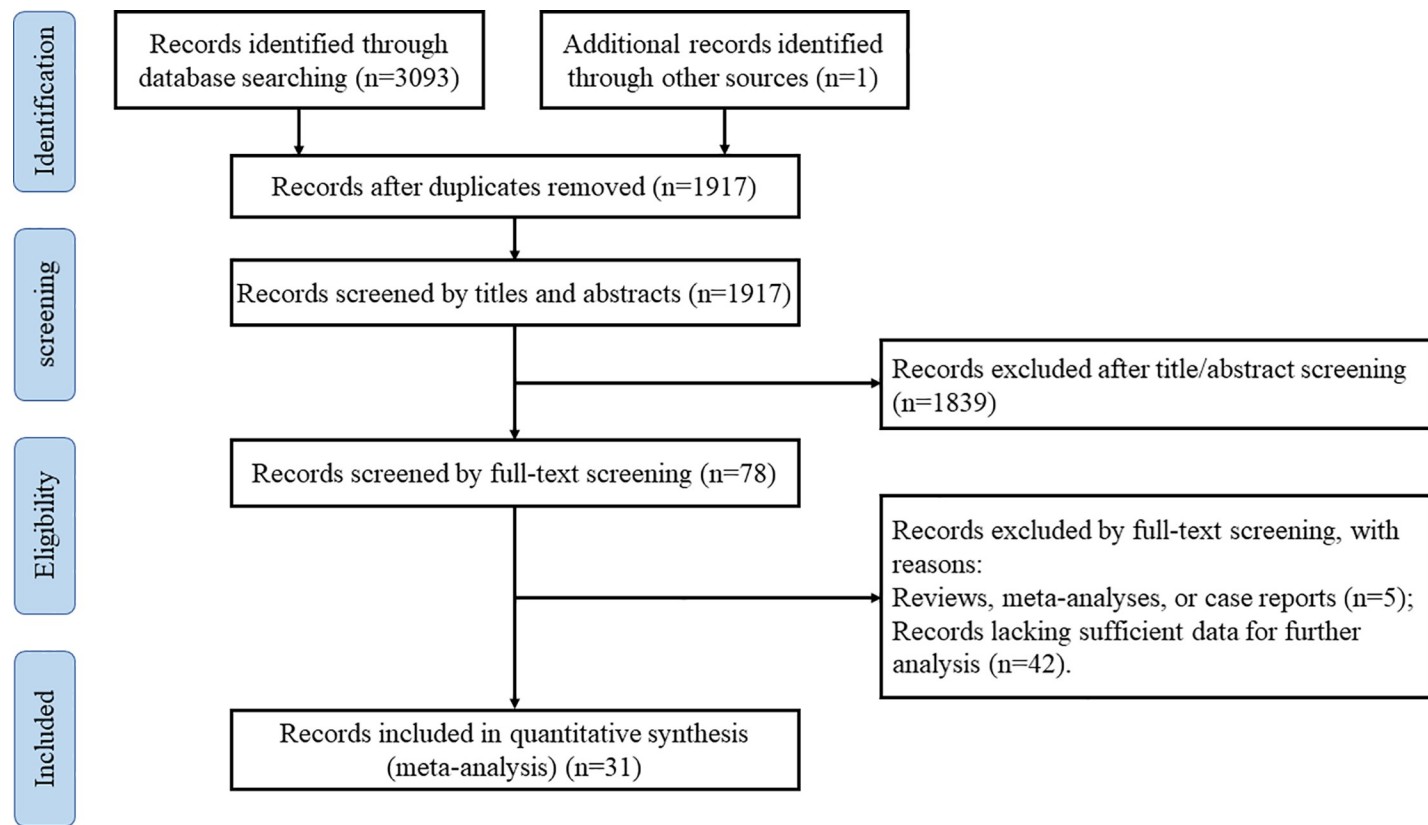

**Fig 1. Flow chart of the literature search.**

## Meta-analysis results of clinical characteristics

Twenty-six studies involving 7274 COVID-19 patients (5926 survivors and 1348 non-survivors) provided data regarding clinical characteristics (**S2 Table**). The association between various clinical characteristics and the risk of mortality in COVID-19 patients were shown in **Fig 2**. Compared with survivors, non-survivors were more likely to present with dyspnea (66% vs. 34%), chest tightness (46% vs. 30%), hemoptysis (4% vs. 3%), expectoration (42% vs. 32%) and fatigue (50% vs. 44%) (**S2 Table**). In addition, dyspnea, chest tightness, hemoptysis, expectoration and fatigue were observed as significant poor risk factors of mortality (dyspnea: OR = 4.52, 95%CI [3.15, 6.48], P<0.001; chest tightness: OR = 2.50, 95%CI [1.78, 3.52], P<0.001; hemoptysis: OR = 2.00, 95%CI [1.02, 3.93], P = 0.045; expectoration: OR = 1.52, 95% CI [1.17, 1.97], P = 0.002; and fatigue: OR = 1.27, 95%CI [1.09, 1.48], P = 0.003). The heterogeneity test results of dyspnea, chest tightness, hemoptysis, expectoration and fatigue evaluated by $I^2$ were 79%, 2%, 0%, 51% and 10%, respectively. However, no significant relationships were found between mortality and fever, cough, diarrhea, headache, abdominal pain, dizziness, nausea, chest pain and so on (**Fig 2**).

## Meta-analysis results of laboratory findings

A total of twenty-four studies consisting of 5900 cases (4639 survivors and 1261 non-survivors) reported laboratory findings of COVID-19 patients (**S3 Table**). We compared the pretreatment absolute leukocytes count, absolute leukocytes count, LDH, D-Dimer, PCT and ferritin between survivors and non-survivors. Compared with survivors, significant increases were found in non-

**Table 1. Characteristics of all included studies.**

| Author | Year | Country | City | MINORS score | Total S | Total NS | Age S | Age NS | Male (%) S | Male (%) NS | Hypertension (%) S | Hypertension (%) NS | Diabetes (%) S | Diabetes (%) NS | Malignancy (%) S | Malignancy (%) NS | CCD (%) S | CCD (%) NS | CB (%) S | CB (%) NS | CRD (%) S | CRD (%) NS | COPD (%) S | COPD (%) NS |
|---|---|---|---|---|---|---|---|---|---|---|---|---|---|---|---|---|---|---|---|---|---|---|---|---|
| Cao JL | 2020 | China | Wuhan | 18 | 85 | 17 | 53 (47, 66) | 72 (63, 81) | 47 | 77 | 20 | 65 | 6 | 35 | 4 | 6 | 2 | 18 | 4 | 18 | 1 | 18 | NA | NA |
| Chen RC | 2020 | China | Guangzhou | 19 | 445 | 103 | 53.5 (13.9) | 66.9 (12.1) | 55 | 67 | 23 | 44 | 9 | 19 | 3 | 3 | NA | NA | 2 | 8 | 3 | 2 | 0 | 5 |
| Chen RC (2) | 2020 | China | Guangzhou | 21 | 1540 | 50 | 48 (1–94) [a] | 69 (51–86) [a] | 57 | 78 | 16 | 56 | 8 | 26 | 1 | 6 | NA | NA | 2 | 12 | 1 | 10 | 1 | 12 |
| Chen T | 2020 | China | Wuhan | 21 | 161 | 113 | 51 (37, 66) | 68 (62, 77) | 55 | 74 | 24 | 48 | 14 | 21 | 1 | 4 | 4 | 14 | 0 | 4 | 1 | 4 | NA | NA |
| Deng Y | 2020 | China | Wuhan | 18 | 116 | 109 | 40 (33, 57) | 69 (62, 74) | 44 | 67 | 16 | 37 | 8 | 16 | 2 | 6 | 3 | 12 | NA | NA | NA | NA | NA | NA |
| Du RH | 2020 | China | Wuhan | 19 | 158 | 21 | 56 (13.5) | 70.2 (7.7) | 55 | 48 | 29 | 62 | 17 | 29 | 2 | 5 | NA | NA | NA | NA | NA | NA | NA | NA |
| Fan H * | 2020 | China | Wuhan | 19 | 26 | 47 | 46.2 (12) | 65.5 (9.7) | 65 | 68 | 12 | 45 | NA | NA | NA | NA | NA | NA | NA | NA | NA | NA | NA | NA |
| Goicoechea M | 2020 | Spain | Madrid | 18 | 25 | 11 | 69 (14) | 75 (6) | 68 | 55 | 100 | 91 | 68 | 55 | NA | NA | NA | NA | NA | NA | NA | NA | 32 | 9 |
| Giacomelli A | 2020 | Italy | Milan | 19 | 185 | 48 | NA | NA | 34 | 19 | NA | NA | NA | NA | NA | NA | NA | NA | NA | NA | NA | NA | NA | 19 |
| Huang J | 2020 | China | Yichang | 18 | 283 | 16 | 52.5 (16.6) | 69.2 (9.7) | 53 | 69 | 22 | 69 | 11 | 25 | 2 | 25 | NA | NA | 4 | 13 | NA | NA | 2 | 19 |
| Javanian M | 2020 | Iran | Babol | 18 | 81 | 19 | 57.7 (13.6) | 69.3 (11.1) | 49 | 57 | 25 | 63 | 33 | 53 | 1 | 16 | 15 | 42 | 1 | 11 | 9 | 26 | 9 | 26 |
| Li LL | 2020 | China | Wuhan | 19 | 68 | 25 | 43.7 (13.1) | 69 (10.5) | 38 | 60 | 0 | 20 | 9 | 20 | 4 | 4 | 0 | 16 | NA | NA | NA | NA | 9 | 8 |
| Nowak B | 2020 | Poland | Warsaw | 18 | 123 | 46 | 59.3 (20.1) | 75.3 (11.9) | 46 | 65 | 43 | 59 | 13 | 35 | 16 | 33 | 29 | 48 | NA | NA | 22 | 17 | 11 | 20 |
| Ruan QR | 2020 | China | Wuhan | 19 | 82 | 68 | 50 (44, 81) | 67 (15, 81) | 65 | 72 | 28 | 43 | 16 | 18 | 1 | 3 | 0 | 19 | 6 | 10 | 0 | 3 | 1 | 3 |
| Shi Q | 2020 | China | Wuhan | 21 | 259 | 47 | NA | NA | 47 | 60 | 38 | 68 | NA | NA | 4 | 9 | 12 | 36 | 3 | 15 | 3 | 11 | NA | NA |
| Shi SB * | 2020 | China | Shanghai | 21 | 609 | 62 | 61 (49, 70) | 74 (66, 81) | 47 | 57 | 27 | 60 | 13 | 27 | 3 | 7 | NA | NA | 2 | 13 | 3 | 19 | 3 | 3 |
| Sun H | 2020 | China | Wuhan | 18 | 123 | 121 | 67 (64, 72) | 72 (66, 78) | 42 | 68 | 50 | 63 | 20 | 23 | NA | NA | NA | NA | NA | NA | NA | NA | NA | 3 |
| Tang N | 2020 | China | Wuhan | 19 | 162 | 21 | 52.4 (15.6) | 64 (20.7) | 51 | 76 | NA | NA | NA | NA | NA | NA | NA | NA | NA | NA | NA | NA | NA | NA |
| Wang DW | 2020 | China | Wuhan | 19 | 88 | 19 | 44.5 (35, 58.8) | 73 (64, 81) | 47 | 84 | 18 | 53 | 7 | 26 | NA | NA | 7 | 37 | 3 | 16 | 2 | 5 | 2 | 5 |
| Wang K | 2020 | China | Wuhan | 21 | 470 | 78 | 58 (46–67) [a] | 67 (61.8–78) [a] | 48 | 71 | 27 | 49 | 14 | 24 | 5 | 4 | NA | NA | NA | NA | 1 | 6 | 2 | 9 |
| Wang L (1) [1] | 2020 | China | Wuhan | 18 | 274 | 65 | 68 (64, 74) | 76 (70, 83) | 46 | 60 | 39 | 50 | 16 | 17 | 4 | 5 | 12 | 33 | 4 | 16 | 3 | 6 | 4 | 17 |
| Wang L (2) | 2020 | China | Wuhan | 19 | 169 | 33 | 61 (49, 67) | 74 (65, 84) | 39 | 70 | 26 | 49 | 11 | 12 | 4 | 6 | 7 | 15 | 2 | 21 | 3 | 12 | 2 | 15 |
| Wang Y * | 2020 | China | Nanjing | 19 | 211 | 133 | 57 (47–69) | 70 (62–77) | 50 | 56 | 34 | 52 | 16 | 23 | NA | NA | 9 | 17 | NA | NA | NA | NA | 1 | 10 |
| Wu CM * | 2020 | China | Wuhan | 19 | 40 | 44 | 50 (40.3, 56.8) | 68.5 (59.3, 75) | 78 | 66 | 18 | 36 | 13 | 25 | NA | NA | 10 | 9 | NA | NA | NA | NA | NA | NA |
| Xu PP | 2020 | China | MC | 21 | 659 | 33 | 45 (14.6) | 64.7 (13.4) | 53 | 73 | 14 | 52 | 7 | 36 | 1 | 3 | 3 | 36 | NA | NA | 1 | 9 | 1 | 12 |
| Xu B | 2020 | China | Wuhan | 21 | 117 | 28 | 56 (43, 66) | 73 (68, 77.3) | 50 | 61 | 18 | 36 | NA | NA | NA | NA | NA | NA | NA | NA | 2 | 7 | NA | NA |
| Yang XB * | 2020 | China | Wuhan | 19 | 20 | 32 | 51.9 (12.9) | 64.6 (11.2) | 70 | 66 | NA | NA | 10 | 22 | 5 | 3 | 10 | 9 | 0 | 22 | NA | NA | NA | NA |
| Yang KY | 2020 | China | MCr | 21 | 165 | 40 | 62 (57, 69) | 63 (53, 75) | 41 | 73 | 34 | 28 | 12 | 5 | NA | NA | NA | NA | NA | NA | NA | NA | 3 | 0 |
| Yan XS | 2020 | China | Wuhan | 19 | 964 | 40 | 62 (50, 70) | 68 (58, 79) | 48 | 68 | 22 | 50 | 10 | 25 | 1 | 3 | 0 | 38 | 2 | 23 | NA | NA | 1 | 0 |
| Zhang J * | 2020 | China | Wuhan | 18 | 11 | 8 | 68 (38, 87) | 77 (66, 91) | 55 | 63 | 55 | 63 | 9 | 38 | NA | NA | NA | NA | 9 | 25 | NA | NA | NA | NA |
| Zhou F | 2020 | China | Wuhan | 21 | 137 | 54 | 52 (45, 58) | 69 (63, 76) | 59 | 70 | 23 | 48 | 14 | 32 | 2 | 0 | NA | NA | NA | NA | 0 | 4 | 2 | 7 |

Abbreviation: COVID-19, coronavirus disease 2019; S: survivors; NS: non-survivors; CCD: Chronic cardiac disease; CB: Cerebrovascular disease; CRD: Chronic renal disease; COPD: Chronic obstructive pulmonary disease; MINORS: Methodological Index for Non-Randomized Studies; NA, Not available; MC: Multi-center.

a: Reported as median (range). Other studies were reported as median (IQR) or mean (SD).

*: All patients with ARDS or severe/critically ill patients.

[1]: All patients were over 60 years old.

| Clinical manifestation | No. of Patients | No. of Research | OR(95 %CI) | $I^2$ | P Value |
|---|---|---|---|---|---|
| Dyspnea | 6556 | 21 | 4.52 (3.15, 6.48) | 79% | <0.001 |
| Chest tightness | 904 | 3 | 2.50 (1.78, 3.52) | 2% | 0.359 |
| Hemoptysis | 1476 | 6 | 2.00 (1.02, 3.93) | 0% | 0.517 |
| Expectoration | 4455 | 14 | 1.52 (1.17, 1.97) | 51% | 0.014 |
| Fatigue | 5912 | 20 | 1.27 (1.09, 1.48) | 10% | 0.333 |
| Anorexia | 1708 | 8 | 1.53 (0.99, 2.37) | 56% | 0.026 |
| Dizziness | 1513 | 6 | 1.29 (0.85, 1.98) | 0% | 0.702 |
| Chest pain | 845 | 4 | 1.14 (0.62, 2.10) | 41% | 0.164 |
| Fever | 7192 | 26 | 1.06 (0.88, 1.27) | 0% | 0.852 |
| Nausea | 1557 | 5 | 1.01 (0.59, 1.74) | 45% | 0.123 |
| Cough | 7146 | 25 | 0.93 (0.73, 1.18) | 57% | 0.000 |
| Emesis | 1629 | 6 | 0.89 (0.54, 1.49) | 37% | 0.159 |
| Headache | 5076 | 14 | 0.84 (0.61, 1.14) | 9% | 0.353 |
| Myalgia | 3826 | 15 | 0.83 (0.66, 1.05) | 0% | 0.706 |
| Diarrhea | 6290 | 19 | 0.82 (0.67, 1.01) | 0% | 0.474 |
| Pharyngalgia | 2294 | 8 | 0.73 (0.44, 1.19) | 26% | 0.224 |
| Abdominal pain | 1965 | 6 | 0.71 (0.39, 1.31) | 0% | 0.683 |
| Shiver | 2086 | 3 | 0.68 (0.37, 1.23) | 3% | 0.359 |

-0.5  0.5  1.5  2.5  3.5  4.5  5.5  6.5

**Fig 2. Meta-analysis results of the relationship between clinical manifestation and the increasing risk of mortality in COVID-19 patients.** Abbreviation: OR, odds ratio; CI, confidence interval.

survivors in pretreatment absolute leukocytes count (WMD = $3.27 \times 10^9$/L, 95% CI [2.34, 4.21], P<0.001) (Table 2, S1 Fig) and we further observed significant negative correlation between the risk of mortality and decreased pretreatment absolute leukocytes count (OR = 0.32, 95%CI [0.22, 0.46], P<0.001; $I^2$ = 44%, P = 0.11) (Fig 3). The mean value of pretreatment absolute lymphocytes count was significantly decreased in non-survivors with a WMD of $-0.39 \times 10^9$/L, 95% CI [-0.46, -0.33]; P<0.001) compared with survivors (Table 2, S1 Fig) and the reduction of pretreatment absolute lymphocytes count was also significantly related to the increased risk of mortality (OR = 9.83, 95%CI [6.72, 14.38], P<0.001). No pronounced heterogeneity was observed by the heterogeneity test ($I^2$ = 0%, P = 0.515) (Fig 3). What's more, LDH, D-Dimer, PCT and ferritin were also found to be elevated in non-survivors (Table 2, S1 Fig) and the increased indicators mentioned above were also associated with increased risk of mortality (Fig 3).

## Publication bias

The funnel plots and Egger's tests showed that there was no evidence of publication bias either in any clinical characteristic analysis or in any laboratory test analysis (S2 and S3 Figs).

## Discussion

In this article, we summarized the incidence of some common symptoms of COVID-19 infections and found that dyspnea, chest tightness, hemoptysis, expectoration and fatigue were

**Table 2. Meta-analysis results of comparing laboratory abnormalities between survivor and non-survivor COVID-19 patients.**

| Laboratory findings | No. of the studies | No. of the patients | WMD | P | Test of heterogeneity $I^2$ (%) P | |
|---|---|---|---|---|---|---|
| Leukocytes ($\times 10^9$/L) | 19 | 5408 | 3.27 (2.34, 4.21) | <0.001 | 90 | <0.001 |
| Lymphocytes ($\times 10^9$/L) | 20 | 4825 | -0.39 (-0.46, -0.33) | <0.001 | 83 | <0.001 |
| **Lactate dehydrogenase (LDH) (U/L)** | 13 | 3336 | 211.60 (148.63, 274.57) | <0.001 | 68 | 0.008 |
| **Procalcitonin (ng/mL)** | 11 | 3330 | 0.31 (0.20, 0.42) | <0.001 | 88 | <0.001 |
| **D-Dimer (μg/mL)** | 17 | 3108 | 4.97 (3.55, 6.39) | <0.001 | 90 | <0.001 |
| **Ferritin (ng/mL)** | 6 | 1500 | 770.05 (530.34, 1009.76) | <0.001 | 86 | <0.001 |

Abbreviation: WMD, weighted mean difference; LDH, lactate dehydrogenase.

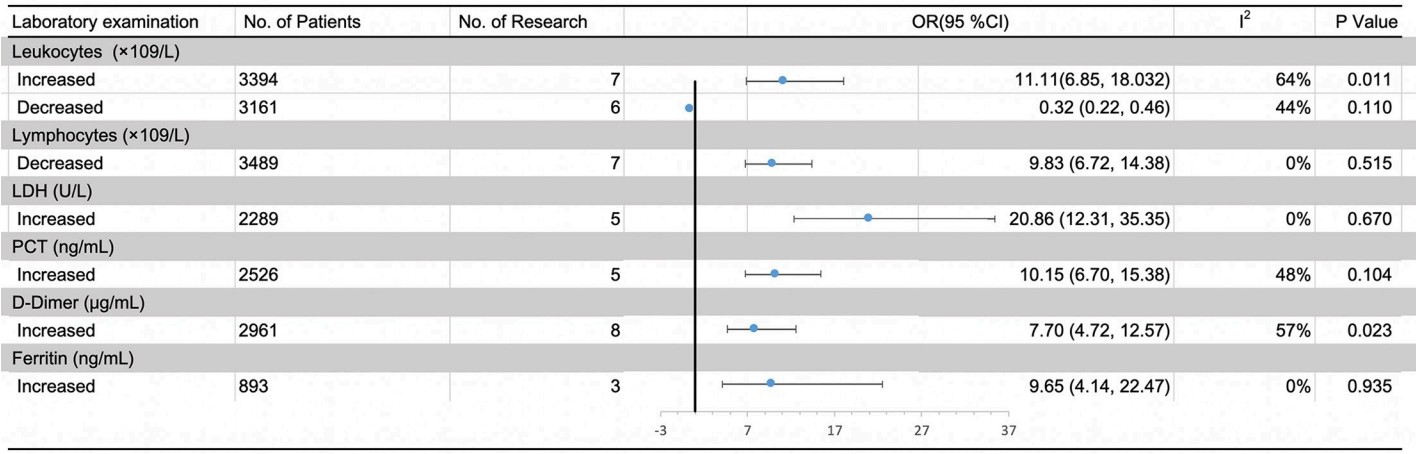

| Laboratory examination | No. of Patients | No. of Research | OR(95 %CI) | I² | P Value |
|---|---|---|---|---|---|
| Leukocytes (×109/L) | | | | | |
| Increased | 3394 | 7 | 11.11(6.85, 18.032) | 64% | 0.011 |
| Decreased | 3161 | 6 | 0.32 (0.22, 0.46) | 44% | 0.110 |
| Lymphocytes (×109/L) | | | | | |
| Decreased | 3489 | 7 | 9.83 (6.72, 14.38) | 0% | 0.515 |
| LDH (U/L) | | | | | |
| Increased | 2289 | 5 | 20.86 (12.31, 35.35) | 0% | 0.670 |
| PCT (ng/mL) | | | | | |
| Increased | 2526 | 5 | 10.15 (6.70, 15.38) | 48% | 0.104 |
| D-Dimer (μg/mL) | | | | | |
| Increased | 2961 | 8 | 7.70 (4.72, 12.57) | 57% | 0.023 |
| Ferritin (ng/mL) | | | | | |
| Increased | 893 | 3 | 9.65 (4.14, 22.47) | 0% | 0.935 |

**Fig 3. Meta-analysis results of the relationship between laboratory abnormalities and the increasing risk of mortality in COVID-19 patients.** Abbreviation: OR, odds ratio; CI, confidence interval.

significantly associated with poor prognosis in COVID-19 patients. For laboratory tests, our study indicated significant increased pretreatment absolute leukocytes count and decreased pretreatment absolute lymphocytes count were observed in non-survivors and they were also associated with the increased risk of mortality in COVID-19 patients.

As an emerging infectious disease, the rapid global rise of COVID-19 pneumonia infections and deaths has attracted significant attention. To foresee the prognosis of COVID-19 infected individuals, it is essential to ascertain the risk factors for death fast and reliably. A large number of clinical studies have explored the clinical characteristics and laboratory examinations of severe and critical COVID-19 patients. Zheng et al. [36] reported that the fever, shortness of breath or dyspnea indicated the disease deterioration. Our results were consistent with the findings of Shi et al. that the presence of dyspnea was risk factors for death, rather than fever [37]. Another recent retrospective study of 179 patients with confirmed COVID-19 found that fatigue and expectoration were more frequently observed in non-survivors than survivors, which were associated with increased risk of mortality [9]. Hemoptysis was an uncommon symptom in COVID-19 patients [38]. In several studies, the incidence of hemoptysis was higher in survivors [9, 14], while many others reported that hemoptysis occurred more often in non-survivors [6, 7, 17], consistent with our observations. More researches on the role of hemoptysis in predicting the prognosis of COVID-19 patients was required.

For laboratory tests, in addition to pretreatment absolute leukocytes and lymphocytes count, increased LDH, PCT, and ferritin were also observed in non-survivors. Further analyses showed them were all associated with the mortality of patients.

Concerning lymphocyte, some studies found no significant correlation between lymphocyte counts and the severity of the disease [39, 40], whereas other research concluded that lymphopenia was a good predictor of disease progression [41, 42]. The present study is the first meta-analysis, which identified the correlation between lymphopenia and mortality in COVID-19 patients. Regardless of the baseline disease severity, lymphocyte was significantly lower on admission and maintained a lower level during hospitalization in non-survivors, while it increased after treatment in survivors [6–8, 43, 44]. The lymphopenia may result from destruction of lymphocytes (particularly T lymphocytes) and suppression of the proliferation of lymphocytes caused by virus invasion, and recovered lymphocyte could be a predictor of gradual recovery [45].

The present study had some limitations that should be acknowledged. First, all included studies were retrospective. Secondly, subgroup analyses were not performed due to the limited data we can draw from the enrolled studies. Additionally, due to the limitations of language, we included the studies written in English and Chinese only.

## Conclusions

To sum up, we found that dyspnea, chest tightness, hemoptysis, expectoration and fatigue were predictors of increased risk of mortality. Besides, significantly increased pretreatment absolute leukocyte count, PCT, D-Dimer, LDH and ferritin, and decreased pretreatment absolute lymphocyte count were identified in non-survivors, which were all related to increased risk of mortality. Motoring these indicators during the hospitalization of patients plays a very important role in predicting the prognosis of patients. Collectively, our results are helpful in clinical practice, which should be verified by additional large-sample or multi-center studies.

## Supporting information

**S1 Fig.** Forest plot of the laboratory abnormalities (A) leukocytes, (B) lymphocytes, (C) lactate dehydrogenase (LDH), (D) procalcitonin, (E) D-Dimer, (F) ferritin levels in survivors versus non-survivors.
(TIF)

**S2 Fig.** The publication bias of the clinical characteristics (A. dyspnea; B. chest tightness; C. hemoptysis; D. expectoration; E. fatigue; F. anorexia; G. dizziness; H. chest pain; I. fever; J. nausea; K. cough; L. emesis; M. headache; N. myalgia; O. diarrhea; P. pharyngalgia; Q. abdominal pain; R. shiver) between survivors and non-survivors.
(TIF)

**S3 Fig.** The publication bias of the laboratory abnormalities (A) increased leukocytes, (B) decreased leukocytes, (C) decreased lymphocytes, (D) increased lactate dehydrogenase (LDH), (E) increased procalcitonin (PCT), (F) increased D-Dimer, (G) increased ferritin between survivors and non-survivors.
(TIF)

**S1 Table. The results of the quality assessment for each individual study.**
(XLSX)

**S2 Table. Clinical characteristics of survivor and non-survivor COVID-19 patients.** Abbreviation: CI, confidence interval; NA, not available.
(XLSX)

**S3 Table. Laboratory abnormalities of survivor and non-survivor COVID-19 patients.** Abbreviation: CI, confidence interval; SD, standard deviation.
(XLSX)

**S1 File. Prisma-2009-checklist.**
(DOC)

**S2 File. Search strategy for meta-analysis of risk factors for predicting mortality of COVID-19 patients (PubMed via NLM).**
(DOCX)

## Author Contributions

**Conceptualization:** Jing Jin, Bojiang Chen, Weimin Li.

**Data curation:** Lan Yang.

**Formal analysis:** Lan Yang.

**Funding acquisition:** Wenxin Luo, Weimin Li.

**Methodology:** Jing Jin, Wenxin Luo, Yuncui Gan.

**Software:** Lan Yang, Jing Jin, Wenxin Luo, Yuncui Gan.

**Supervision:** Bojiang Chen, Weimin Li.

**Validation:** Yuncui Gan.

**Visualization:** Wenxin Luo.

**Writing – original draft:** Lan Yang, Jing Jin, Wenxin Luo, Yuncui Gan.

**Writing – review & editing:** Bojiang Chen, Weimin Li.

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
