## [Decision Letter · Decision Letter 0]

27 Oct 2020

PONE-D-20-30258

Risk factors for predicting mortality of COVID-19 patients : A systematic review and meta-analysis

PLOS ONE

Dear Dr. Li,

Thank you for submitting your manuscript to PLOS ONE. After careful consideration, we feel that it has merit but does not fully meet PLOS ONE’s publication criteria as it currently stands. Therefore, we invite you to submit a revised version of the manuscript that addresses the points raised during the review process.

The reviewers have commented on your above paper. They have suggested that this manuscript be revised according to the reviewers suggestions and resubmitted.  Provided you address the changes recommended, the manuscript will be accepted for publication

We look forward to receiving your revised manuscript.

Kind regards,

Prof. Raffaele Serra, M.D., Ph.D

Academic Editor

PLOS ONE

Journal Requirements:

2. Please attach a Supplemental file of the results of the quality assessment for each individual study assessed, reporting the outcome for each individual criteria considered.

3. Please include the date(s) on which you accessed the databases or records to obtain the data used in your study.

4. Please provide a citation for the MINORS score.

5.Thank you for stating the following in the Funding Section of your manuscript:

[This work was supported by National Nature Science Foundation of China [grant

227 numbers 91859203 and 81871890] and Major Science and Technology Innovation

228 Project of Chengdu City [grant number 2020-YF08-00080-GX]]

 [The author(s) received no specific funding for this work.]

6. We suggest you thoroughly copyedit your manuscript for language usage, spelling, and grammar. If you do not know anyone who can help you do this, you may wish to consider employing a professional scientific editing service.  

Additional Editor Comments (if provided):

The reviewers have commented on your above paper. They have suggested that this manuscript be revised according to the reviewers suggestions and resubmitted.

Reviewers' comments:

Reviewer's Responses to Questions

**Comments to the Author**

1. Is the manuscript technically sound, and do the data support the conclusions?

Reviewer #1: Yes

2. Has the statistical analysis been performed appropriately and rigorously? 

Reviewer #1: Yes

3. Have the authors made all data underlying the findings in their manuscript fully available?

Reviewer #1: Yes

4. Is the manuscript presented in an intelligible fashion and written in standard English?

Reviewer #1: Yes

5. Review Comments to the Author

Reviewer #1: The authors aimed to perform a systematic meta-analysis to summarize the clinical characteristics and laboratory test before treatment among COVID-19 patients 65 and identify the possible risk factors for mortality. The article is timely and novel and it is overall well structured and written.

Nevertheless, I would improve the manuscript focusing also on cardiovascular disease that increases poor prognosis and related mortality. For this purpose read and cite the article by Ielapi N, et al. Cardiovascular disease as a biomarker for an increased risk of COVID-19 infection and related poor prognosis. Biomark Med. 2020 Jun;14(9):713-716.

6. PLOS authors have the option to publish the peer review history of their article (what does this mean?). If published, this will include your full peer review and any attached files.

Reviewer #1: No

---

## [Author Response · Author response to Decision Letter 0]

6 Nov 2020

Dear Editor-in-Chief and reviewers:

Thank you for your letter and for the reviewers' comments concerning our manuscript entitled “Risk factors for predicting mortality of COVID-19 patients: A systematic review and meta-analysis”. All suggestions were very helpful for us to revise and improve our paper. We carefully studied these comments and made corrections that we hope meet with approval. The revised portions are marked with ‘Track changes’ in the manuscript.

Here are my responses to the Editor-in-Chief’ comments.

Response: We have now re-formatted our paper carefully to meet PLOS ONE’s style requirements.

2. Please attach a Supplemental file of the results of the quality assessment for each individual study assessed, reporting the outcome for each individual criteria considered.

Response: The results of the quality assessment for each individual study were presented in S1 Table. We have added S1 table in the revised version of our manuscript. We change the original “eTable 1” to “S2 Table” and the original “eTable 2” to “S3 Table”. We are sorry for making some mistakes in calculating the MINORS scores. After re-calculating all scores of enrolled studies, the MINORS score of Chen T(2020) was changed from 18 to 21, the MINORS score of Goicoechea M (2020) was changed from 21 to 18, and the MINORS score of Zhou F (2020) was changed from 18 to 21.(Page 8-10, Table 1)

3. Please include the date(s) on which you accessed the databases or records to obtain the data used in your study.

Response: We conducted a systematic search in PubMed, Scopus, Web of Science and Embase to identify studies in patients with COVID-19 infection up to 4 June 2020. This was mentioned in “Materials and methods”-“Search strategy”. (Page 4, Line 64).

4. Please provide a citation for the MINORS score.

Response: The citation for the MINORS score was provided as reference [3]. (Page 5, Line 88)

Slim K, Nini E, Forestier D, Kwiatkowski F, Panis Y, Chipponi J. Methodological index for non-randomized studies (MINORS): development and validation of a new instrument. ANZ Journal of Surgery. 2003;73(9):712-6.

5.Thank you for stating the following in the Funding Section of your manuscript:

[This work was supported by National Nature Science Foundation of China [grant

227 numbers 91859203 and 81871890] and Major Science and Technology Innovation

228 Project of Chengdu City [grant number 2020-YF08-00080-GX]]

 [The author(s) received no specific funding for this work.]

Response: We have removed any funding-related text from the manuscript and add the information of funding in cover letter.

6. We suggest you thoroughly copyedit your manuscript for language usage, spelling, and grammar. If you do not know anyone who can help you do this, you may wish to consider employing a professional scientific editing service.

Response: The full manuscript has been reviewed and edited by a professional scientific English editor.

Here are my responses to the reviewers’ comments.

Reviewer #1: The authors aimed to perform a systematic meta-analysis to summarize the clinical characteristics and laboratory test before treatment among COVID-19 patients 65 and identify the possible risk factors for mortality. The article is timely and novel and it is overall well structured and written.

Nevertheless, I would improve the manuscript focusing also on cardiovascular disease that increases poor prognosis and related mortality. For this purpose read and cite the article by Ielapi N, et al. Cardiovascular disease as a biomarker for an increased risk of COVID-19 infection and related poor prognosis. Biomark Med. 2020 Jun;14(9):713-716.

Response: Thank you for reviewing our manuscript and your advices were helpful. According to your suggestion about the impact of cardiovascular disease on COVID-19 infection and prognosis. We did analysis to detect the relationship between hypertension and chronic cardiac disease and mortality of COVID-19. The results showed that hypertension (OR= 2.94, 95%CI [2.39, 3.62], P<0.001) and chronic cardiac disease (OR= 3.89, 95%CI [2.65, 5.72], P<0.001), were also associated with increased mortality of COVID-19. The detail information was provided in the following table.

　 No. of the studies No. of the patients OR, 95%CI P-value Heterogeneity

 I2 P-value

Hypertension 28 8939 2.94 [2.39, 3.62] <0.001 54.90% <0.001

CCD 17 3806 3.89 [2.65, 5.72] <0.001 53.40% 0.005

We appreciate the editor/reviewers' earnest work and hope that the corrections will make the revised manuscript acceptable for publication. Once again, thank you very much for your comments and suggestions, and we look forward to hearing from you.

---

## [Editor Report · Decision Letter 1]

17 Nov 2020

Risk factors for predicting  mortality of COVID-19 patients : A systematic review and meta-analysis

PONE-D-20-30258R1

Dear Dr. Li,

We’re pleased to inform you that your manuscript has been judged scientifically suitable for publication and will be formally accepted for publication once it meets all outstanding technical requirements.

Kind regards,

Prof. Raffaele Serra, M.D., Ph.D

Academic Editor

PLOS ONE

Additional Editor Comments (optional):

amended manuscript is acceptable.
---

## [Editor Report · Acceptance letter]

19 Nov 2020

PONE-D-20-30258R1 

Risk factors for predicting mortality of COVID-19 patients: A systematic review and meta-analysis

Dear Dr. Li:

I'm pleased to inform you that your manuscript has been deemed suitable for publication in PLOS ONE. Congratulations! Your manuscript is now with our production department. 

Kind regards, 

on behalf of

Prof. Raffaele Serra 

Academic Editor

PLOS ONE